# Gait Disturbance in Patients with Schizophrenia in Relation to Walking Speed, Ankle Joint Range of Motion, Body Composition, and Extrapyramidal Symptoms

**DOI:** 10.3390/healthcare13060604

**Published:** 2025-03-10

**Authors:** Ryuichi Tanioka, Reiko Kamoi, Yoshihiro Mifune, Keita Nakagawa, Kaito Onishi, Krishan Soriano, Hidehiro Umehara, Hirokazu Ito, Leah Bollos, Rick Yiu Cho Kwan, Kyoko Osaka, Mai Sato, Eiji Takigawa, Kyoko Goda, Hironari Kamoi, Takeru Ishii, Shoko Edo, Kazushi Mifune, Tetsuya Tanioka

**Affiliations:** 1Faculty of Health Sciences, Hiroshima Cosmopolitan University, Hiroshima 731-3166, Japan; tanioka@hcu.ac.jp (R.T.); nakagawa@hcu.ac.jp (K.N.); 2Mifune Hospital, Marugame 763-0073, Japan; reihamlintel@gmail.com (R.K.); ymifune29@gmail.com (Y.M.); big.i.animal.8g@gmail.com (E.T.); godafamily.1002@gmail.com (K.G.); worldismine512@icloud.com (H.K.); i.takeru0209@icloud.com (T.I.); edo3yoro49@icloud.com (S.E.); mifune@mifune-hp.jp (K.M.); 3Graduate School of Health Sciences, Tokushima University, Tokushima 770-8509, Japan; kai10.onishi10@gmail.com (K.O.); ksoriano@spup.edu.ph (K.S.); leahaclb@gmail.com (L.B.); 4Graduate School of Biomedical Sciences, Tokushima University, Tokushima 770-8509, Japan; sanntyoumenoumehara@gmail.com (H.U.); h.itoh@tokushima-u.ac.jp (H.I.); 5School of Nursing, Tung Wah College, Hong Kong, China; rickkwan@twc.edu.hk; 6Department of Nursing, Nursing Course of Kochi Medical School, Kochi University, Kochi 783-8505, Japan; osaka@kochi-u.ac.jp; 7Department of Rehabilitation, Shikoku Chuo Medical and Welfare Academy, Shikoku Chuo 799-0422, Japan; maaaichi1285@yahoo.co.jp

**Keywords:** gait rehabilitation, gait disturbance, gait speed, extrapyramidal symptoms, drug-induced extrapyramidal symptoms scale (DIEPSS), antipsychotic dosage, range of motion, ankle plantar flexion, muscle mass, bone mineral content

## Abstract

Background/Objectives: In patients with schizophrenia, gait disturbances (e.g., reduced walking speed and stride length) are linked to neural dysfunction and extrapyramidal symptoms. To inform gait rehabilitation strategies, this study examines the relationships of walking speed with extrapyramidal symptoms, stride length, antipsychotic dosage, ankle joint range of motion, and body composition in patients with chronic schizophrenia. Methods: Sixty-eight patients with chronic schizophrenia were included. All variables were described based on their measurement levels using non-parametric methods. Spearman’s rho was calculated to assess correlations. For multiple linear regression analyses, backward stepwise elimination was used to determine variables associated with walking speed. Statistical significance was set to *p* < 0.05. Results: Walking speed was positively correlated with stride length, chlorpromazine-equivalent dose, ankle plantar flexion, body mass index, bone mineral content, trunk muscle mass, and skeletal muscle mass index. In contrast, it was negatively correlated with drug-induced extrapyramidal symptoms scale (DIEPSS) scores for gait, bradykinesia, tremor, overall severity, and age. The multiple linear regression indicated that DIEPSS 2 bradykinesia level and ankle plantar flexion angle, adjusted for a 26% variance, best explained the walking speed. Conclusions: A lower bradykinesia severity and a higher ankle plantar flexion are associated with higher walking speeds. Thus, it is critical to assess stride length, bradykinesia, angle/limitation/torque of ankle plantar flexion, trunk and upper and lower limb muscle masses, and walking speed in patients with chronic schizophrenia. Specific strategies for gait rehabilitation should focus on stride training, plantar flexion strengthening exercises, and balance training.

## 1. Introduction

Dysfunction of the frontoparietal, subcortical [1], prefrontal [2], and cerebellar [3] networks contributes to gait disturbances in individuals with schizophrenia. A key characteristic of gait disturbances in schizophrenic patients is a reduced walking speed [4,5], which is further exacerbated by stride length dysregulation [6]. Gait disturbances in patients with schizophrenia can be associated with parkinsonism, such as stiffness, tremor, and shuffling gait, all of which result from drug-induced extrapyramidal symptoms (EPSs) [7,8]. Patients taking typical antipsychotics are at greater risk of developing EPSs than those taking atypical antipsychotics [9]. Parkinsonism may lead to an abnormal forward flexion of the trunk, neck, and extremities [10], such an abnormal flexion being linked to coordination and sensory integration disorders in the limbs, further exacerbating gait disturbances [11]. Moreover, the forward-leaning posture in patients with parkinsonism weakens the abdominal muscles [12,13], affects factors such as knee flexion or extension related to ankle plantar flexion [14], restricts the plantar flexion of the ankle joint, and impairs the body’s balance [15,16,17].

The trunk muscles are integral to maintaining balance and walking speed [18]. However, patients with schizophrenia experience higher rates of sarcopenia and dynapenia [19,20], as well as reduced muscle mass [21]. The use of antipsychotic medication is associated with a decrease in bone mineral density (BMD) and worsening physical impairments [22]. Therefore, patients with schizophrenia are often treated with physical therapy to prevent secondary physical illnesses, complications, and lifestyle-related diseases [23]. However, only a few studies have focused on the relationship of walking speed with ankle joint range of motion (ROM), body composition, EPSs, and antipsychotic medication dosage in patients with schizophrenia.

This study aims to examine the relationships of walking speed with EPSs, stride length, antipsychotic dosage, ankle joint ROM, and body composition in patients with chronic schizophrenia.

This study tests two hypotheses: (1) gait parameters, functioning, EPSs, antipsychotic dosage, and body composition are correlated in individuals with schizophrenia, and (2) walking speed is associated with EPSs, body composition, and ankle joint ROM.

## 2. Materials and Methods

### 2.1. Study Design

This cross-sectional study, which followed the Strengthening the Reporting of Observational Studies in Epidemiology (STROBE) guidelines [24], was conducted from 18 May 2024 to 30 November 2024.

### 2.2. Setting

Participants were recruited at Mifune Hospital. It is a 328-bed hospital with psychiatry, internal medicine, dentistry, and oral surgery departments.

### 2.3. Participants

The target patients were asked by their treating psychiatrists to participate in the study. Eligible patients who fulfilled the criteria mentioned below were invited to participate. This group of patients with chronic schizophrenia comprised the same cohort reported in a previous publication [20]. A total of 173 patients were enrolled, consented, and participated in the study. However, 105 patients were excluded from the analyses because these patients were unable to undergo measurements of walking speed or body composition (*N* = 91) or blood tests (*N* = 92) owing to severe mental symptoms, such as delusions. Finally, data from 68 patients with complete sets of test results were analyzed.

#### 2.3.1. Inclusion Criteria

(1) Aged 28–86 years; (2) diagnosed with schizophrenia according to the Diagnostic and Statistical Manual of Mental Disorders 5 (DSM-5) criteria [25]; (3) hospitalized for more than one year since illness onset and admitted to a “psychiatric care ward requiring long-term treatment” under the Japanese medical fee system; and (4) received treatment and exhibited stable symptoms.

#### 2.3.2. Exclusion Criteria

(1) Diagnosed with severe mental disorders other than schizophrenia according to the DSM-5 criteria, such as delusional disorders, bipolar II or related disorders, depressive disorders, neurodevelopmental disorders, substance-related or addictive disorders, or personality disorders; (2) catatonic schizophrenia; (3) unable to understand instructions due to a medical condition or medication status; (4) disabling medical diseases (e.g., cerebrovascular disease, stroke, neurological disease, and cancer); and (5) patients who were unable to complete the full set of measurements (gait speed, ankle ROM, and body composition assessment) due to severe psychiatric symptoms.

### 2.4. Variables and Measurements

This study measured several variables related to gait, ankle joint ROM, EPSs, psychotropic drug use, global assessment of functioning, and body composition.

#### 2.4.1. Gait

Gait measurements included walking speed and stride length when the patients were instructed to perform a 6 m walk test [26] on an NEC gait analysis system (NEC Corporation, Tokyo, Japan) [27].

#### 2.4.2. ROM—Ankle Joints

The maximum painless plantar/dorsiflexion ROM of the ankle joints was measured using a goniometer while the patient was seated [28].

#### 2.4.3. Extrapyramidal Symptoms

The drug-induced extrapyramidal symptoms scale (DIEPSS) [29] is a practical multidimensional rating scale that assesses drug-induced, unwanted extrapyramidal effects and quantifies the severity of objectively observed symptoms. The DIEPSS includes eight distinct items assessed through objective observation on a five-point scale, where 0 represents normal behavior and 4 represents a severe behavioral abnormality. This scale comprises eight individual parameters (gait, bradykinesia, sialorrhea, muscle rigidity, tremor, akathisia, dystonia, and dyskinesia) and one global assessment. The global item, overall severity, considers the severity and frequency of individual items, subjective distress, and their influences on daily activities [30].

#### 2.4.4. Psychotropic Drugs

The recorded atypical antipsychotics included olanzapine, risperidone, aripiprazole, ziprasidone, clozapine, amisulpride, quetiapine fumarate, and paliperidone extended-release tablets. The daily dosages of these antipsychotic medications were converted to chlorpromazine-equivalent daily doses per 100 mg according to international consensus [31,32,33]. The daily doses of different medications can be compared by converting the dosage of the above medications to equivalents.

#### 2.4.5. Functioning

The global assessment of functioning (GAF) [34] was used to evaluate the psychological, social, and occupational functioning of the participants. Disruptions to functionality owing to physical or environmental constraints were not considered. The clinician evaluated the patient with a score between 1 and 100 for the current or previous period, with higher scores indicating better functioning. The GAF has been reported to be reliable and valid for measuring psychiatric disturbances in patients with severe mental illnesses [35].

### 2.5. Body Composition Assessment

An RD-545 InnerScan Pro (RD-545 InnerScan Pro; TANITA Corporation. Tokyo, Japan) provided an in-depth analysis of 26 body composition variables [36]. Measurements included weight, body fat, muscle mass, muscle quality score, body mass rating, bone mineral content (estimated weight of bone minerals in the body), visceral fat level, basal metabolic rate, metabolic age, total body water, and BMI. This device can measure fat and muscle individually in the arm, leg, and trunk segments. Visceral fat accumulation was indicated by a visceral fat level score [37]. The total limb skeletal muscle mass index (SMI) (kg) was calculated from the information obtained from the body mass data, and the data were divided by the square of the corresponding height (m^2^).

### 2.6. Study Size Estimation

The sample size was calculated using G*Power version 3.1.9.7 (Heinrich-Heine Universität Düsseldorf, Düsseldorf, Germany) for multiple linear regression analysis. The test family included F-tests, with a statistical test of multiple linear regression with a fixed model, an R^2^ deviation from zero, an effect size f^2^ of 0.35, an error probability α of 0.05, a power of 0.95, and predictors of 3 [38]. The required sample size for this study was >54.

### 2.7. Statistical Methods

The variables were first described based on their measurement levels. The Shapiro–Wilk’s test was used to analyze normality. Owing to the small sample size, non-parametric methods were used to describe the variables (i.e., frequencies with percentages for categorical variables and medians with ranges for continuous variables).

To test hypothesis #1, Spearman’s rho was calculated for walking speed, stride length, EPSs, chlorpromazine-equivalent dose (mg/day), ROM of the plantar and dorsiflexion angles of the ankle joint, age, BMI, bone mineral content, and muscle mass.

To test hypothesis #2, multiple linear regression models were used to determine whether the variables were independently associated with walking speed. For the regression, backward stepwise elimination was used to identify variables associated with walking speed.

All statistical analyses were performed using the Jamovi statistical software version 2.4.11.0 (The Jamovi PPleroject, Sydney, Australia) [39]. Statistical significance was set to *p* < 0.05.

## 3. Results

### 3.1. Descriptive Data

Descriptive data for all variables are shown in Table 1. The median (min–max) age of the participants was 64 (28–86) years, disease duration was 38 (1 to 70) years, the GAF score was 25 (5–35) points, the BMI was 22.35 (13.4–34.1) kg/m^2^, the height was 160 (139–177) cm, the body weight was 56.5 (31.0–94.9) kg, the bone mineral content was 2.2 (1.4–3.1) kg, the SMI was 6.52 (4.83–8.83) kg/m^2^, and the trunk muscle mass was 20.05 (14.5–31.8) kg.

The walking speed was 0.683 (0.0167–1.28) m/s. The right- and left-foot stride lengths were 43 (2–70) and 40 (0–71) cm, respectively. The chlorpromazine-equivalent dose was 629 (12.6–2382) mg/day. The plantar flexion angles of the right and left ankle joints during automatic motion were 30 (0 to 75) and 32.5 (−10 to 75) degrees, respectively. The dorsiflexion angles of the right and left ankle joints during automatic motion were 17.5 (−10 to 45) and 15 (−15 to 45) degrees, respectively.

The median walking speeds and percentages of participants by the DIEPSS 9 overall severity category are provided in Table 1. The male-to-female ratio of the study participants was 1:1.

### 3.2. Main Results

Table 2 presents the correlations for each measure. A significant strong positive correlation was found between walking speed and stride length (Rt, ρ = 0.83, *p* < 0.001; Lt, ρ = 0.88, *p* < 0.001). Significant positive correlations were also found between walking speed and chlorpromazine-equivalent dose (ρ = 0.26, *p* < 0.05), ankle plantar flexion (Rt, ρ = 0.37, *p* < 0.01; Lt, ρ = 0.41, *p* < 0.001); BMI (ρ = 0.23, *p* < 0.05), bone mineral content (ρ = 0.35, *p* < 0.01), trunk muscle mass (ρ = 0.27, *p* < 0.05), and SMI (ρ = 0.25, *p* < 0.05). In contrast, a significant negative strong correlation was found between walking speed and age (ρ = 0.61, *p* < 0.001).

As shown in Table 3, walking speed was negatively correlated with DIEPSS 1 gait (*p* < 0.001), DIEPSS 2 bradykinesia (*p* < 0.001), DIEPSS 5 tremor (*p* < 0.05), and DIEPSS 9 overall severity (*p* < 0.05).

In the univariate analyses, walking speed was positively correlated with stride length (Rt, ρ = 0.83, *p* < 0.001; Lt, ρ = 0.88, *p* < 0.001), chlorpromazine-equivalent dose (*p* < 0.05), and ankle plantar flexion (Rt, ρ = 0.37, *p* < 0.01. Lt, ρ = 0.41, *p* < 0.001) but negatively correlated with DIEPSS 1 gait (*p* < 0.001) and DIEPSS 2 bradykinesia (*p* < 0.001).

The backward stepwise elimination returned the following results:(1)When all variables with *p* < 0.01 were included in the univariate analysis, the intercept was significant (t = −2.23, *p* = 0.03) and the significant variables were right and left stride lengths (both *p* < 0.001) and DIEPSS 2 bradykinesia (*p* < 0.05).(2)When the right and left stride lengths were removed from 1), the intercept was significant (t = 6.4, *p* < 0.001); however, no significant variables were found.(3)When the right stride length, ankle plantar flexion, DIEPSS 1 gait, and DIEPSS 2 bradykinesia were entered, the intercept was not significant (t = −0.35, *p* = 0.73). Similarly, when the left stride length, ankle plantar flexion, DIEPSS 1 gait, and DIEPSS 2 bradykinesia were entered, the intercept was also not significant (t = −0.73, *p* = 0.47).(4)When the right and left stride lengths were removed, the intercepts were significant for the right (t = 6.67, *p* < 0.001) and left (t = 6.85, *p* < 0.001), and ankle plantar flexion was significant for the right (t = 3.13, *p* < 0.001) and left (t = 3.45, *p* < 0.001).

Finally, the variables listed in Table 4 best explained the relationships with walking speed. The results of the multiple linear regression analysis are as follows. On the right side, ankle plantar flexion (*p* < 0.001) and DIEPSS 2 bradykinesia levels (level 2, *p* < 0.05; level 3, *p* < 0.01) were significantly associated, with an adjusted R^2^ coefficient of 0.259. On the left side, ankle plantar flexion (*p* < 0.001) and DIEPSS 2 bradykinesia level (level 2, *p* < 0.05; level 3, *p* < 0.01) were significant variables, and the adjusted R^2^ coefficient was 0.273.

## 4. Discussion

**Hypothesis** **1.***Gait parameters, functioning, EPSs, antipsychotic dosage, and body composition are correlated in individuals with schizophrenia*.

The key findings of this study were as follows. Gait speed had a strong positive correlation with stride length. Furthermore, walking speed was significantly positively related to the chlorpromazine-equivalent dose, ankle plantar flexion, BMI, bone mineral content, trunk muscle mass, and SMI. In contrast, a negative correlation was found between walking speed and age, DIEPSS 1 gait, DIEPSS 2 bradykinesia, DIEPSS 5 tremor, and DIEPSS 9 overall severity scores.

A correlation was observed between walking speed and ankle joint plantar flexion angle. A previous study indicated that enhancing the ankle plantar flexion strength and velocity can lead to an increased walking speed [40]. Moreover, an increase in velocity is associated with a reduction in the dorsiflexion angle, whereas plantar flexion angles show an increase [41].

The reference values for ankle joint ROMs were set to 0 to 20 degrees for ankle dorsiflexion and 0 to 45 degrees for ankle plantar flexion, as established by the Japanese Society of Rehabilitation Medicine [42]. In this study, some patients exhibited a ROM limitation of −10 to 15 degrees in the right ankle joint and −10 to 15 degrees in the left ankle joint, as well as a hypermobility of approximately 20 degrees. However, the normal ROM varies individually and is affected by factors such as age, sex, and physical activity levels. Several studies have reported an overall ROM in the sagittal plane between 65 and 75 degrees, including 10 to 20 degrees of dorsiflexion and 40 to 55 degrees of plantar flexion [28]. Moreover, the reference standard and the measurements taken in the seated position differ on average by approximately 20 degrees, indicating that ankle joint ROMs may not be accurately measured when seated [42]. However, in the current study, we measured all participants in a seated position to account for factors such as patient refusal. This approach may have affected the measured ROM values.

The standard value for normal walking speed is approximately 1.0 m/s [43], and a value of <0.8 m/s is considered a high-risk factor for falling [44]. Although the stride length varies with age and sex, it is generally considered to be approximately 60 cm in women and 70 cm in men [45]. GAF scores indicated poor mental health, and the median walking speed of the participants was 0.683 m/s. The right and left stride lengths were 43 (2–70) and 40 (0–71) cm, respectively, indicating that the maximum stride lengths of both legs were close to the reference value (60–70, women and men), although the stride lengths were narrower than this reference value.

Using the DIEPSS 9 overall severity category, the median walking speed was 0.83 m/s for level 0, 0.64 m/s for level 1, 0.63 m/s for level 2, and 0.37 m/s for level 3, indicating a decrease in walking speed at all levels. EPSs can lead to small steps or a festinating gait. An increase in pitch was anticipated to enhance gait speed even with a reduced stride length.

However, since cadence and stride length are the main determinants of gait speed [46], the relationship between cadence and stride length as an indicator of gait abnormalities by EPSs needs to be investigated in future studies.

In this study, the chlorpromazine-equivalent dose was 629 mg. A dose equivalent to 200–300 mg of chlorpromazine is considered the minimum effective dose, whereas doses exceeding 1000 mg are considered high [47]. Although the motor system is not directly affected by schizophrenia, pharmacogenic EPSs caused by adverse effects of pharmacological therapies—often the primary focus of treatment—frequently arise [8,9]. At first glance, the finding that patients taking high doses of antipsychotics walked faster is surprising. However, as EPSs become more pronounced, physical function declines, the patient ages, and the dosage is usually reduced; therefore, this finding is consistent with clinical reality.

Our analysis revealed positive relationships between walking speed and BMI, bone mineral content, trunk muscle mass, and SMI. The study population included patients with a wide age range (28–86 years). The BMI values varied from underweight to obese. Additionally, bone mineral content, SMI, and trunk muscle mass were below the normative values in some individuals.

The SMI in our study population was 6.52 kg/m^2^. An SMI below 7.0 kg/m^2^ in men and 5.7 kg/m^2^ in women indicates sarcopenia [48]. According to a patient survey conducted by the Ministry of Health, Labour and Welfare, the length of hospitalization for schizophrenia in Japan often exceeds one year [49]. Prolonged hospitalization can result in physical complications and reduced mobility. Furthermore, reports indicate that patients with schizophrenia tend to have lower overall endurance and agility [50] compared to healthy individuals. Reports indicate that walking can enhance bone density by introducing a suitable weight to the foot. Specifically, it is advisable to engage in brisk walking three or more times per week for at least 30 min per session [51].

Literature reviews indicate that factors such as inadequate physical activity, malnutrition, smoking, alcohol intake, and insufficient vitamin D levels contribute to reduced BMD and an increased risk of osteoporosis [52]. Additionally, bone loss is linked to muscle loss, and older individuals with low bone density frequently exhibit low muscle mass [53,54]. Vancampfort et al. [55] reported that in patients with schizophrenia, impaired walking capacity compared to age-, sex-, and BMI-matched healthy controls was associated with reduced health-related muscular fitness. Hayashida et al. [56] reported that the relationship between muscle strength and muscle mass differs according to sex and age. These findings suggest that muscle strength is different from muscle mass and that an individualized approach to prevent the decline of muscle strength and muscle mass is necessary for health promotion in patients with chronic schizophrenia.

Based on the above discussion, it was assumed that the critical variables related to walking speed were ankle plantar flexion angle, DIEPSS 9 overall severity, DIEPSS 1 gait, DIEPSS 2 bradykinesia, DIEPSS 5 tremor, BMI, bone mineral content, trunk muscle mass, SMI, and age. Thus, multiple linear regression models were examined with these variables.

**Hypothesis** **2.**
*Walking speed is associated with EPSs, body composition, and ankle joint ROM.*


The key findings of this study are as follows. In the multiple linear regression analyses, bradykinesia, as assessed by the DIEPSS, and the magnitudes of the right/left ankle plantar/dorsiflexion angles had the highest explanatory power for gait speed, with an adjusted R^2^ of approximately 26%. In other words, less severe bradykinesia and larger ankle joint plantar/dorsiflexion angles were associated with faster walking speeds. However, the multiple linear regression model did not include body composition variables.

Martin et al. [5] reported that a forward-leaning posture was the most common postural feature in the early and late stages of schizophrenia and discussed the association between postural changes and disease severity. These authors also noted that the patients’ decreased walking speed was due to a shortened stride length [5]. Reduced gait speed due to parkinsonism in patients with schizophrenia is a major factor that increases the risk of falls [57]. The relationship among gait speed, EPSs, antipsychotic dosage, and ankle joint ROM shown in this study is important for assessing fall risk. Particularly, the adjustment of the antipsychotic medication dose is important for fall prevention [58]. In addition, rehabilitation to maintain and expand ankle joint function contributes to the stabilization of gait speed, which, in turn, reduces the risk of falls [59]. Patients with schizophrenia and parkinsonism may also avoid certain activities if they experience falls. In addition, predictors of repeated falls in schizophrenia are frailty, physical function, cognitive function, and sex [60].

As walking speed varies, the functions of the ankle joint adapt accordingly: during collision work, energy is absorbed when the foot meets the ground; in rebound work, energy is reused after the impact; in preload work, energy is stored for the subsequent propulsive force; and in push-off work, energy is produced when the foot lifts off the ground to create propulsion. During push-off, energy is produced as the foot departs from the ground to facilitate propulsion. Through this series of processes, the ankle joint adapts to changes in gait speed and plays an important role in efficient energy utilization and generation of propulsive force [61]. As the gait speed increases, the role of the ankle joint becomes more important. According to Hu et al. [61], the ankle and femoral joints absorb more mechanical energy and play a role in impact mitigation as the gait speed increases. The ankle joint also generates energy to increase propulsion and is responsible for pushing the distal segment forward in cooperation with the femoral joint. This significantly alters the function of the ankle joint to accommodate changes in gait speed. In other words, the ankle joint plays various roles in absorbing shock, storing energy, and generating a propulsive force which may act as a mechanism for responding to changes in walking speed.

Therefore, ankle plantar flexion, which affects the heel strike during gait, has an important influence on gait speed, and gait rehabilitation should focus on ankle softness and ROM.

The ankle plantar flexor muscles are critical for balance control and locomotion. Impairments in the strength of ankle plantar flexors can result in ineffective push-off and reduced walking speed [62]. Our findings examined relationships between walking speed and the ankle plantar flexion angle (static ROM). In the current study, only plantar and dorsiflexion angles of the ankle joint were measured. Future studies should analyze the relationships among plantar flexion ROM, muscle strength, and ankle joint torque.

### 4.1. Interpretation and Generalizability/Implications

The novelty of this study lies in its comprehensive investigation of diverse factors influencing gait speed and their interactions in patients with schizophrenia. Previous studies have primarily examined individual factors such as antipsychotic dosage and EPSs. Our study revealed how multiple factors influence each other by comprehensively analyzing these factors. This reaffirms the importance of physical health in effective physical rehabilitation and treatment strategies for patients with schizophrenia and underscores the need for a comprehensive approach.

### 4.2. Limitations and Future Studies

A limitation of this study is the small sample size. As a result, it was difficult to ensure sufficient statistical power in the analysis, and the data did not reflect the characteristics of the patients who refused assessment due to severe psychiatric symptoms. Future studies should consider improving the measurement method and providing a measurement environment in which patients can relax and reduce their psychological burden.

In this study, the maximum painless plantar/dorsiflexion ROM of the ankle joints was measured using a goniometer while the patient was seated. In addition, some patients in the study population had limitations in ankle dorsiflexion. Therefore, it is important to consider physical rehabilitation for patients with chronic schizophrenia. These limitations exist because ankle ROM was not measured during walking. In the future, it will be necessary to analyze the movement of the ankle joint, knee joint, and lower limbs.

In future studies, the use of the weight-bearing lunge test [63,64] to assess dorsiflexion may provide a more accurate representation of ankle mobility during gait. In addition, incorporating dynamic assessments of plantar flexion during actual gait, rather than relying solely on static ROM measurements, would provide more relevant insights into ankle function under real-world conditions. This approach can provide a more comprehensive understanding of ankle mobility and its impact on gait in individuals with schizophrenia.

Quantitative measures of gait unsteadiness [65] may be useful in assessing gait speed and stride as well as EPSs in patients with chronic schizophrenia. Also, cadence and stride length as indicators of gait abnormalities caused by EPSs need to be analyzed in future studies.

## 5. Conclusions

This study aims to examine the relationships among walking speed, EPSs, stride length, antipsychotic dosage, ankle joint ROM, BMI, muscle mass, and bone mineral contents in patients with schizophrenia to inform gait rehabilitation strategies. This study hypothesizes that (1) gait, EPSs, antipsychotic dosage, functioning, and body composition are correlated in people living with schizophrenia and (2) walking speed is associated with EPSs, body composition, and ankle joint ROM.

In line with hypothesis #1, a strong correlation was identified between walking speed and stride length. Significant relationships were also observed between walking speed and chlorpromazine-equivalent dose, ankle plantar flexion, BMI, bone mineral content, trunk muscle mass, and SMI. In contrast, walking speed was negatively correlated with measurements related to age, DIEPSS 1 gait, DIEPSS 2 bradykinesia, DIEPSS 5 tremor, and DIEPSS 9 overall severity.

Regarding hypothesis #2, multiple linear regression analyses showed that bradykinesia, as assessed using the DIEPSS, and the magnitude of the right/left ankle plantar/dorsiflexion angles had the highest explanatory power to explain gait speed, with an adjusted R^2^ coefficient of approximately 26%.

Thus, it is critical to assess stride length, bradykinesia, angle/limitation/torque of ankle plantar flexion, trunk and upper and lower limb muscle masses, and walking speed in patients with chronic schizophrenia. Specific strategies for gait rehabilitation should focus on stride training, plantar flexion strengthening exercises, and balance training.

## Figures and Tables

**Table 1 healthcare-13-00604-t001:** Descriptive data for all variables (*N* = 68).

Variable	Median (Range)
Age, years	64 (28 to 86)
Disease duration, years	38 (1 to 70)
Global assessment of functioning (GAF), points	25 (5 to 35)
Body mass index, kg/m^2^	22.35 (13.4 to 34.1)
Height, cm	160 (139 to 177)
Weight, kg	56.5 (31.0 to 94.9)
Bone mineral content, kg	2.2 (1.4 to 3.1)
Skeletal muscle index, kg/m^2^	6.52 (4.83 to 8.83)
Trunk muscle mass, kg	20.05 (14.5 to 31.8)
Walking speed, m/s	0.683 (0.0167 to 1.28)
Right stride, cm	40 (0 to 71)
Left stride, cm	43 (2 to 70)
Chlorpromazine-equivalent dose, mg/day	629 (12.6 to 2382)
Right ankle plantar flexion, degree	30 (0 to 75)
Left ankle plantar flexion, degree	32.5 (−10 to 75)
Right ankle dorsiflexion, degree	17.5 (−10 to 45)
Left ankle dorsiflexion, degree	15 (−15 to 45)
	Percentage (%), Walking speed, median (min–max)
Drug-induced extrapyramidal symptoms scale 9, overall severity	
0 = None, normal	21 (30.9), 0.83 (0.017–1.28)
1 = Very mild, uncertain	18 (26.5), 0.64 (0.033–1.15)
2 = Mild	26 (38.2), 0.63 (0.07–1.28)
3 = Moderate	3 (4.4), 0.37 (0.33–0.63)
Sex	
Male	34 (50.0)
Female	34 (50.0)

**Table 2 healthcare-13-00604-t002:** Relationships of walking speed with stride length, extrapyramidal symptoms, antipsychotic dose, ankle plantar and dorsiflexion angles, age, and body composition (*N* = 68).

	Stride (Right)	Stride (Left)	DIEPSS 9, Overall Severity	Chlorpromazine-Equivalent Dose	Ankle Plantar Flexion (Right)	Ankle Plantar Flexion (Left)	Ankle Dorsiflexion (Right)	Ankle Dorsiflexion (Left)	Age (Years)	BMI (kg/m^2^)	Bone Mineral Content (kg)	Muscle Mass Trunk (kg)	SMI (kg/m^2^)
Walking speed (m/s)	0.83 ***	0.88 ***	−0.28	0.26 *	0.37 **	0.41 ***	−0.1	−0.13	−0.61 ***	0.23 *	0.36 **	0.27 *	0.25 *

Note. * *p* < 0.05, ** *p* < 0.01, *** *p* < 0.001, one-tailed. BMI: body mass index; DIEPSS: drug-induced extrapyramidal symptoms scale; SMI: limb skeletal muscle mass index.

**Table 3 healthcare-13-00604-t003:** Results of univariate analyses correlating walking speed with extrapyramidal symptoms (*N* = 68).

	DIEPSS 1 Gait	DIEPSS 2 Bradykinesia	DIEPSS 3 Sialorrhea	DIEPSS 4 Muscle Rigidity	DIEPSS 5 Tremor	DIEPSS 6 Akathisia	DIEPSS 7 Dystonia	DIEPSS 8 Dyskinesia	DIEPSS 9 Overall Severity
Walking speed (m/s)	−0.51 ***	−0.42 ***	−0.17	−0.12	−0.29 *	−0.09	−0.13	−0.15	−0.28 *

Note. * *p* < 0.05, *** *p* < 0.001. DIEPSS: drug-induced extrapyramidal symptoms scale.

**Table 4 healthcare-13-00604-t004:** Coefficients of multiple linear regression models describing the associations of walking speed with ankle plantar flexion and extrapyramidal symptoms.

			95% CI							
Predictor	Estimate	SE	Lower	Upper	t	*p*	Stand. Estimate	R	R^2^	Adjusted R^2^
Intercept ^a^	0.626	0.079	0.468	0.784	7.92	<0.001		0.551	0.303	0.259
Ankle plantar flexion (Right)	0.007	0.002	0.003	0.01	3.49	<0.001	0.396			
DIEPSS 2 bradykinesia:										
1—0	−0.157	0.079	−0.315	0.002	−1.98	0.052	−0.496			
2—0	−0.214	0.095	−0.403	−0.025	−2.26	0.027	−0.676			
3—0	−0.41	0.125	−0.66	−0.16	−3.27	0.002	−1.297			
Intercept ^a^	0.627	0.08	0.44	0.76	7.33	<0.001		0.563	0.317	0.273
Ankle plantar flexion (Left)	0.006	0.002	0.003	0.009	3.69	<0.001	0.379			
DIEPSS 2 bradykinesia:										
1—0	−0.152	0.078	−0.322	−0.006	−1.94	0.057	−0.57			
2—0	−0.199	0.094	−0.405	−0.027	−2.11	0.039	−0.684			
3—0	−0.343	0.121	−0.596	−0.104	−2.83	0.006	−1.106			

^a^ Represents the reference level. CI: confidence interval; DIEPSS: drug-induced extrapyramidal symptoms scale; SE: standard error.

## Data Availability

The data presented in this study are available upon request from the corresponding author. The data are not publicly available due to privacy and ethical restrictions.

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
