# Peer review of "Gait Disturbance in Patients with Schizophrenia in Relation to Walking Speed, Ankle Joint Range of Motion, Body Composition, and Extrapyramidal Symptoms"

_healthcare, 2025, doi:10.3390/healthcare13060604_

Round 1

Reviewer 1 Report

Comments and Suggestions for Authors

Greetings.

First of all, thanks for your work. The main theme it's really interesting and important to better care to this kind of patients. The main structure of a paper was followed and it's well written, easy to read and clear. The introductions delivers enough information about existing knowledge. The methods are well choose and the statistical approach well done. The discussion made clear the importance of the results.

I just have a small suggestion, in methods, you've referred that study was made with chronic patients, however, there is no inclusion criteria referring to time of diagnosis or admission, so how can you say that was chronic patients?

Best regards

Author Response

Reviewer 1

We thank the Editor and reviewers for considering our manuscript. The constructive comments and valuable suggestions have greatly helped improve the manuscript’s quality. In response to the suggestions, we have made comprehensive revisions. Details of these modifications are outlined in the resubmitted version. Additionally, we have provided a point-by-point response to each comment below, with our replies highlighted in yellow. We look forward to your favorable reply.

Reviewer 1’s comment:

I just have a small suggestion, in methods, you've referred that study was made with chronic patients, however, there is no inclusion criteria referring to time of diagnosis or admission, so how can you say that was chronic patients?

Response: Thank you for identifying this omission. We have added the required information in the “2.3.1. Inclusion Criteria” section on page 3.

Hospitalized for more than one year since illness onset and admitted to a “psychiatric care ward requiring long-term treatment” under the Japanese Medical Fee System

Also, to clarify that the patient has chronic-phase schizophrenia, the disease duration has been added to Table 1 on pages 4 to 5. The median (min–max) disease duration was 38 (1 to 70) years. Thank you for your advice.

Reviewer 2 Report

Comments and Suggestions for Authors

This study investigates how various physical and clinical factors influence walking speed in schizophrenia patients. The strong correlation between gait speed and stride length, as well as associations with EPS severity, antipsychotic dosage, and ankle ROM, highlight the importance of targeted rehabilitation strategies to improve mobility in this population. The research is methodologically sound, employs validated measurement tools, and presents clinically relevant findings. This research has important implications for gait rehabilitation strategies in this population. Below, I provide detailed feedback on various aspects of the manuscript, including clarity, methodology, results interpretation, and potential areas for improvement.

Abstract

- The abstract is well-structured but could be more concise, especially in the background section. For example, the phrase "To inform gait rehabilitation strategies, this study examined the relationships among walking speed, extrapyramidal symptoms, stride length, antipsychotic dosage, ankle joint range of motion, body mass index, muscle mass, and bone mineral content in patients with schizophrenia." can be streamlined for better readability.

- The methods section mentions Spearman’s rho and stepwise regression, but it would be helpful to specify the statistical significance threshold (e.g., p-value).

- Expanding the clinical implications of findings slightly would enhance the impact of the study.

Keywords

- The keywords should align more closely with the core aspects of the study. For example, "Drug-Induced Extrapyramidal Symptoms Scale (DIEPSS)" is mentioned in the abstract but is missing from the keywords. If it is crucial to the study, it should be included.

- Some keywords, such as "plantarflexion of ankle joints", might be too detailed. Using "ankle plantarflexion" or "ankle range of motion" would improve clarity.

Introduction

(Lines 50-52) The sentence can be revised for readability.

→ A key characteristic of gait disturbances in schizophrenia is reduced walking speed [4-6], which is further exacerbated by stride length dysregulation [7].

(Lines 53-54, 57) The terms "parkinsonian-like symptoms" and "parkinsonism" are used interchangeably. To avoid confusion, ensure consistency or briefly define the distinction.

(Lines 59-61) Provide clarification on whether muscle weakness directly leads to plantarflexion restriction or if other mediating factors are involved.

(Lines 74-77) The hypothesis statement can be reworded for better readability.

→ This study tested two hypotheses: (1) gait parameters, EPS, antipsychotic dosage, functioning, and body composition are correlated in individuals with schizophrenia, and (2) walking speed is associated with EPS, body composition, and ankle joint ROM.

Materials and Methods

(Lines 110-111) ROM assessment was conducted with participants seated. Was this the only position considered? If so, does it fully capture functional ROM during gait?

(Lines 125-127) The conversion to chlorpromazine-equivalent daily doses is appropriate, but it would be helpful to explain why this method was chosen over alternative dose standardization approaches.

(Lines 154-157) The use of non-parametric methods is justified due to the sample size. However, a brief explanation of the data distribution (e.g., results of normality tests such as Shapiro-Wilk) would strengthen this choice.

Results

(Lines 170-174) The study initially recruited 173 patients, but 60% (105 participants) were excluded due to severe mental symptoms. This substantial exclusion rate raises concerns about selection bias and the generalizability of the findings. It would be valuable to discuss whether excluded patients differed systematically from those included (e.g., in terms of symptom severity, age, or medication dosage).

Discussion

- The discussion section offers a comprehensive overview of the study's key findings and contextualizes them within existing research. However, its structure could be refined to enhance readability. To achieve this, consider reorganizing the content by first presenting a concise summary of the main findings. Following this, each major factor - such as extrapyramidal symptoms (EPS), antipsychotic dosage, ankle range of motion (ROM), and body composition - should be discussed in a more structured and systematic manner. This approach would not only improve the flow of information but also facilitate easier comprehension of the relationships between various factors and their implications for gait disturbances in schizophrenia patients.

- The discussion of hypotheses could be more explicitly connected to the results rather than simply restating the findings. A critical interpretation of whether the results support or challenge the hypotheses would be beneficial.

- The correlation between walking speed and stride length aligns with prior research. However, the phrase "stride length influences walking speed" should be refined since walking speed is a function of both stride length and cadence. If cadence was measured, it should be discussed. Otherwise, this should be acknowledged as a limitation and suggested for future research.

- The negative correlations with DIEPSS scores align with expectations. Does this suggest that monitoring EPS should be prioritized in gait rehabilitation? Are certain EPS symptoms more predictive of gait dysfunction than others?

- The seated range of motion (ROM) assessment method employed in this study may have resulted in an underestimation of ankle flexibility, potentially limiting the accuracy of the findings. To address this limitation, future studies should consider implementing alternative measurement techniques that more closely mimic functional movements. For instance, utilizing the weight-bearing lunge test for dorsiflexion assessment could provide a more accurate representation of ankle mobility during gait. Furthermore, incorporating dynamic assessments of plantarflexion during actual walking, rather than relying solely on static ROM measurements, would offer more relevant insights into ankle function in real-world conditions. These methodological improvements would enhance the validity and clinical applicability of the results, providing a more comprehensive understanding of ankle joint mobility and its impact on gait in individuals with schizophrenia.

Conclusions

- The statement "physical rehabilitation should be considered in patients with schizophrenia in the chronic phase when performing gait rehabilitation." is too general. Instead, specific rehabilitation strategies should be mentioned (e.g., stride training, strengthening exercises for plantarflexion, balance training).

Comments on the Quality of English Language

There are a few areas where the clarity, conciseness, and consistency of the language could be improved. Proofreading by a native English speaker or professional editor would enhance fluency. Additionally, shortening long sentences and ensuring consistent terminology would further improve readability.

Author Response

Reviewer 2

We thank the Editor and reviewers for considering our manuscript. The constructive comments and valuable suggestions have greatly helped improve the manuscript’s quality. In response to the suggestions, we have made comprehensive revisions. Details of these modifications are outlined in the resubmitted version. Additionally, we have provided a point-by-point response to each comment below, with our replies highlighted in yellow as attachment file. We look forward to your favorable reply.

Reviewer 3 Report

Comments and Suggestions for Authors

Thank you very much for the opportunity to review this very interesting article.

In their work, the authors investigate the gait disturbances in patients with schizophrenia.

Overall, it is a well-written, clear and concise article addressing a key topic.

The reference list is up to date. The supplementary material is appropriate and self-explanatory.

As for the inclusion criteria, the authors could kindly add whether patients with catatonic schizophrenia were excluded from the analysis.

In addition, I would be very much interested to know whether participants were benzodiazepine-free or naïve at the time of the experiment.

Author Response

Reviewer 3

We thank the Editor and reviewers for considering our manuscript. The constructive comments and valuable suggestions have greatly helped improve the manuscript’s quality. In response to the suggestions, we have made comprehensive revisions. Details of these modifications are outlined in the resubmitted version. Additionally, we have provided a point-by-point response to each comment below, with our replies highlighted in yellow. We look forward to your favorable reply.

Reviewer 3’s comment:

Thank you very much for the opportunity to review this very interesting article. In their work, the authors investigate the gait disturbances in patients with schizophrenia. Overall, it is a well-written, clear and concise article addressing a key topic. The reference list is up to date. The supplementary material is appropriate and self-explanatory.

As for the inclusion criteria, the authors could kindly add whether patients with catatonic schizophrenia were excluded from the analysis.

Response: We have added the exclusion criteria for catatonic schizophrenia.

In addition, I would be very much interested to know whether participants were benzodiazepine-free or naïve at the time of the experiment.

Response: Regarding benzodiazepine use, just under half of the patients were taking them at the time of the experiment. Polypharmacy or overdose of benzodiazepines can lead to problems, such as unsteady gait or weakness. In our hospital, however, benzodiazepines are mainly used as sleeping pills, and we use them as monotherapy at the lowest possible dose. Thus, we believe this will have little effect on walking.